# Methodology for Determining the Key Factors for Non-Point Source Management

**Jae Hong Park \*, Jichul Ryu, Dong Seok Shin and Jae Kwan Lee**

Water Environmental Research Department, National Institute of Environmental Research (NIER), Gyeongseo-dong Seo-gu Incheon 22689, Korea
\*   Correspondence: jhong@korea.ac.kr; Tel.: +82-32-560-7670

**Abstract:** Due to the growing significance of water quality degradation by non-point source (NPS) pollution, regions in which NPS management is required should be designated as the management areas. Relevant management measures should be established to control water quality items related to degradation. It is advantageous that the area where the water environment is negatively affected by NPS is provided with legal grounds for NPS management, namely the designation of an NPS management area. This is because if it is designated as an NPS management area, the government can support the budget necessary for the installation of non-point pollution reduction facilities. In order to effectively utilize the limited budget, it is necessary to select and concentrate the area that should be managed first in the NPS management area. For the efficiency of the NPS pollution management within a management region, priority locations or key management sub-regions should be determined to implement differential management plans. Also, in selecting priority management regions, evaluation factors that can reflect the effects of NPS, such as the water quality target excess ratio in the mid-level region (or the total maximum daily load (TMDL) management) which includes the target region (low-level region), the NPS load in land, and non-permeable area ratio, should be quantified and the management order should be defined. Since NPS has local characteristics, the management items should be determined based on turbidity, suspended solid (SS), or total phosphorus (TP) that affect the local water quality. When the water environment is polluted due to non-point pollutants, various materials such as turbidity, SS, TP, *Escherichia coli*, and heavy metals can be set as management items according to local characteristics. However, the most important items to be managed are turbidity, SS, and TP, because if the solid (SS) is present in the water, which is highly turbid and does not sink easily, people can feel unpleasant and feel that the water is not clean, even if they do not analyze the water quality. In addition, in the case of TP, nutrients accumulated in the land are introduced into the river by rainfall, causing eutrophication. People feel uncomfortable because it changes the water color. Other pollutants can only be found to be contaminated after water quality analysis is performed. The water quality target of the management items should be set realistically, based on the situation of the watershed by considering the watershed model, management flow, NPS pollutant reduction plan, the river flow in the management area, and load. All these reflect the characteristics of the region. To evaluate whether the water quality target is achieved after NPS management, a method similar to the one to set the water quality target should be used to review the performance of the management plan. This study introduces specific examples of key factors in establishing an NPS management plan, including consideration factors and methods for the designation of NPS management regions, consideration factors and the selection method for key management areas within a management region, the selection method of management items, the selection method of the water quality target, and an evaluation method of the water quality target.

**Keywords:** non-point source (NPS); management area; water quality target; management item

---

## 1. Introduction

Non-point sources (NPSs) emit water pollutants in unspecified areas, including cities, roads, paddies, forests, and construction sites, in normal situations without rain. These pollutants are sedimented so that they do not drastically change the water quality of rivers. However, during rain, the direct flow to rivers increases, leading to an increase in the NPS load in rivers as the sedimented pollutants in soils are discharged to the water system. This leads to a rapid degradation of water quality and hydro-ecology health threats [1–3].

According to our own investigation, the "non-point source" is called "stormwater pollution", "stormwater", "stormwater runoff", or "stormwater runoff pollution" in the fields of hydraulics and civil engineering, as well as" diffused pollution" in the agriculture field. Also, our keyword analysis used in NPS-related research from 1997 to 2018 via Science Direct shows a frequency of appearance in the following order: non-point source pollution, diffuse pollution, and stormwater.

Compared to point source pollution, NPS water pollution is difficult to prevent, effectively control, and manage. NPS water pollution is caused mainly by rainfall. In particular, there is a huge amount of pollutants during the early period of rainfall, and pollution sources are varied and spread widely through the country. Prevention and control of NPSs is expensive and difficult to perform.

Until now, pollutant treatment has focused on point source pollution, including domestic sewage, industrial wastewater, and agricultural wastewater. With the development of sewage and wastewater treatment technology, those pollutants have drastically reduced. On the other hand, the NPS load control is much less implemented and continues to increase. With continuous urbanization, advanced land use, and the associated increase of the non-permeated surfaces, the NPS load has increased and currently has a great impact on the water environment during rain.

Based on the Biochemical Oxygen Demand (BOD) load, the ratio of the NPS load in 1998 was about 27.0%, and increased to 52.7% in 2003 and 68.3% in 2010, and it is expected to reach 72.1% by 2020 [4,5].

To control NPSs, it is necessary to designate areas susceptible to water damage following rainfall runoffs from NPSs as control regions and to establish relevant control measures. To designate a region as an NPS control region, the effect of the NPS on this region needs to be quantified. Furthermore, the priority in areas within the NPS load control region should be determined for an effective NPS management.

In establishing NPS control measures, the types of water pollutants that require control should be selected. Then, a water quality target should be determined based on the target region, the generation and discharge conditions of water pollutants, and suitable management should be implemented. To perform the control measures, an execution plan should be established and the tasks evaluated regularly to check if the water quality target has been achieved. If not, the NPS management plan needs to be reviewed and issues in the execution plan corrected. And also, any possibility that the water quality target might have been set too low should be reviewed and a plan to achieve the water quality target should be considered.

This study suggests ways to set areas where priority management of NPS is needed, how to determine managed water quality items, and how to set water quality targets for management. Furthermore, a methodology for evaluating the achievement of the target water quality after the management period of the NPS is presented. These results can be used effectively to establish and implement a policy for NPS management.

## 2. Materials and Methods

The NPS management plan targeted 11 regions, designated as NPS management areas until 2018, among 14 regions in 21 local governments.

The management plan for these 11 regions consisted of identical content, evaluation, and diagnosis methods, including the current situation for the management region, the designation rationale, any excess of the water quality standard, the frequency in which the water quality target of the

mid-level region has been exceeded, the management duration, target pollutants, management time required, management target, and the evaluation method of the water quality target.

The water quality target items and region selection were based on the establishment and evaluation of a watershed model suitable for the target region according to each water quality item. Using the daily flow data, a flow duration curve (FDC) was created. If the frequency in which the water quality target was exceeded became very high in the mid-level region (a wider watershed including the target region to which the policy-based water quality target is set) under raining conditions causing NPS pollution so that it needed control, the concerned item was selected as the water quality target item, and the concerned region became a management region.

To select priority management regions within the NPS management region, direct causal factors of the NPS (the water quality target excess ratio in the mid-level region, the NPS load in unit area, the non-permeated area ratio per unit area) were indexed, averaged, and sorted from the highest total score sum.

The management of water quality target used a method based on the load duration curve (LDC) and another based on concentration distribution. The LDC-based method used an FDC to set the water quality target after the NPS load reduction measures. The concentration distribution-based method determined the water quality target in a suitable level of the estimated concentration when NPS pollutants were reduced, according to the optimal scenario within the management flow zone.

The establishment of the water quality of the management target can be considered based on the ecological health of the river, besides the LDC and concentration distribution methods. However, ecological health needed to be considered for various evaluation factors and long-term monitoring and costliness. Therefore, considering the current water quality condition (considering water quality for at least 10 years in this study) and the amount of pollutants generated, the management target water quality could be set at a relatively low cost in a short time. This is because the data that have been monitored since the past could be easily utilized.

Figure 1 shows the flowchart of the whole method.

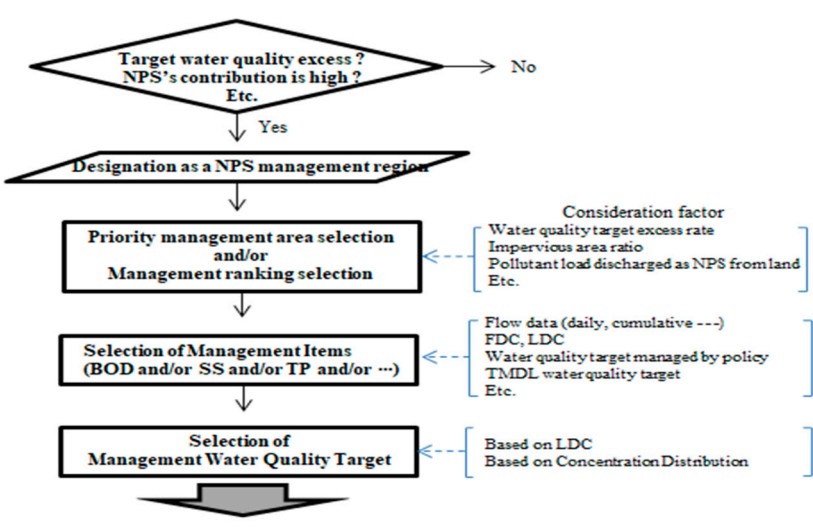

**Figure 1.** Flowchart of the whole method.

As shown in Table 1 and Figure 2, 14 regions in South Korea, including Gwangju Metropolitan City, were designated as NPS management regions [6]. Once a region was designated as NPS management region, a plan to manage NPS was established. The plan included the duration (over 10 years), the target pollutants and their amount, the preventive and reduction measures of the target pollutants, and the management target (usually 90% of the policy water quality standard in the management flow zone).

**Table 1.** Current status of the NPS management regions in Korea.

| | Designated Region | Date | Management Item | Management Flow Zone | Management Water Quality Target | Target Period |
|---|---|---|---|---|---|---|
| 1 | Gwangju Metropolitan city | 2007 | BOD (biochemical oxygen demand) | 5–60% | 5 mg/L | 10 years |
| 2 | Doamho Lake | 2007 | SS (suspended solids) | 5–50% | 5 mg/L | 10 years |
| 3 | Suwon | 2010 | BOD (biochemical oxygen demand) | 5–25% | | 12 years |
| 4 | GoljicheonWatershed | 2013 | SS (suspended solids) | | 90% tile of the SS concentration | 10 years |
| 5 | SaemangeumWatershed | 2013 | TP (total phosphorus) | 5–50% | 90% tile of the TP concentration | 16 years |
| 6 | Upper Soyangho Lake(Mandae Zone) | 2015 | SS (suspended solids) | 10–50% | Estimated water quality concentration in the management flow zone (the water quality concentration with the implementation of the reduction measures using the watershed model) | 10 years |
| 7 | Upper Soyangho Lake (Ga-a Zone) | 2015 | SS (suspended solids) | 10–50% | Estimated water quality concentration in the management flow zone (the water quality concentration with the implementation of the reduction measures using the watershed model) | 10 years |
| 8 | Upper Soyangho Lake (Jawoon Zone) | 2015 | SS (suspended solids) | 10–50% | Estimated water quality concentration in the management flow zone (the water quality concentration with the implementation of the reduction measures using the watershed model) | 10 years |
| 9 | Yangsan City's YangcheonWatershed | 2017 | TP (total phosphorus) | 10–60% | <0.044 mg/L | 13 years |
| 10 | GapcheonWatershed in Daejeon Metropolitan City | 2017 | | | Not specified | |
| 11 | Lower Watershed of AndongDam in Andong | 2018 | TP (total phosphorus) | 5–60% | <0.022 mg/L | 14 years |
| 12 | SeonakdongRiverWatershed in Gimhae | 2018 | TP (total phosphorus) | 5~60% | <0.044 mg/L | 14 years |
| 13 | SingalcheoninYongin, TancheonWatershed | 2018 | | | Not specified | |
| 14 | DaegiZon in SongchoenWatershed, Gangneung | 2018 | | | Not specified | |

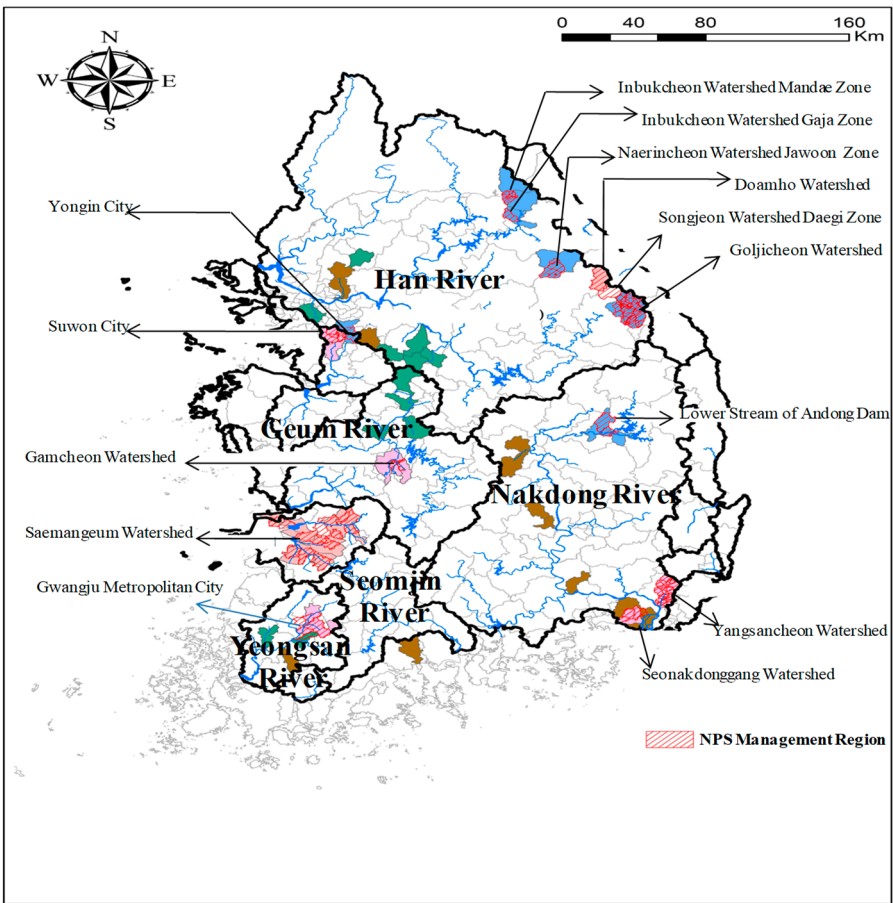

**Figure 2.** Non-point source (NPS) management areas in Korea.

The adoption of an NPS management area means that the area has relatively large non-point pollutants compared to other areas. Therefore, NPS management areas need various measures to reduce the occurrence of non-point pollutants. The contamination of rivers can be prevented through the non-point pollutant reduction of the NPS managed area. In Korea, the Ministry of Environment supports 50–70% of the cost of installing non-point pollution reduction facilities when it is set as an NPS management area. The establishment and management of the NPS management area continuously suppresses the generation of non-point pollutants and contributes to the water quality improvement of the river. In the past case of Gwangju Metropolitan City being designated as an NPS management area, it was evaluated that BOD improved by 27% after the city being designated as an NPS management area.

## 3. Results and Discussion

### 3.1. Preparation of the Management Plan

#### 3.1.1. Designation of Management Regions

When a region has been seriously damaged or seriously affected by damages due to the NPS rainfall runoffs, it is designated and managed as an NPS management region. Damages can include the impossibility of rivers or lakes usage, impacts on the health or assets of residents, or general damages to the national environment. The damage caused by rainfall runoff is that soil in agricultural land is leaked and affects the downstream river. The river is changed into muddy water due to the soil spilled from the farmland (the river is maintained in muddy water for several days due to the minute particles in the soil even after the rainfall is over), and the soil is continuously piled up in the river and damage such as the habitat destruction of aquatic organisms is generated, so dredging of the river sediment is

necessary. Therefore, in order to prevent such damage caused by the NPS rainfall, it is necessary to designate the area as an NPS management area to suppress the occurrence of non-point pollutants.

A region is designated as an NPS management region when its policy water quality target level (the water quality target in the zone) is impaired and the NPS's contribution is high.

Figure 3 shows a case study that analyzed the effect of NPS based on suspended solids (SS) [7]. The data of the Figure 3 were used for the recent 20 years of Yongsan Bridge located in Chuncheon, Gangwon-do. After a suitable watershed model had been established in the target watershed, the daily flow data of previous 20 years were used to create the FDC, which was multiplied to the SS water quality target in the region (25 mg/L) to derive the water quality target LDC. The derived LDC was compared with the result of the SS load simulation in the region, based on the NPS load reduction project, to analyze the frequency of the excessive water quality target. The results show that under the top 25% of the high flow condition, that is, the rainfall condition with water pollution by NPS, the rate of the SS load was high and ranged between 67% and 94%. Therefore, SS in this region is largely affected by NPSs, and the region is designated as an NPS management region to reduce SS.

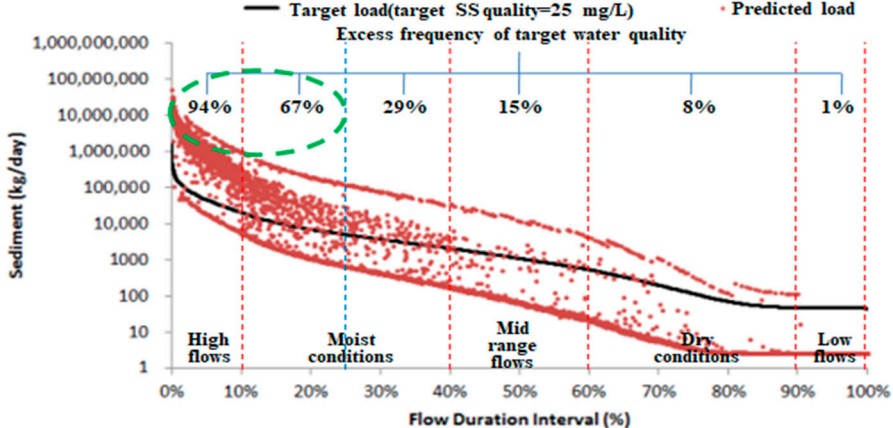

**Figure 3.** Analysis of the effects of non-point pollution sources.

### 3.1.2. Selection of the Key Management Regions

Once designated an NPS management region, locations (regions within the management region) that require concentrated and prioritized management are determined based on water quality and current pollutant load. When determining the priority order, common factors affecting NPSs must be reviewed and objectified by quantitative indexation. In this study, the following three factors were selected and indexed: the level of the water quality target of a specific water quality item set by policy in an NPS management region was exceeded; the pollutant load was discharged as non-point form from land; and the impervious area ratio that increased water pollution by NPS pollutants (the area ratio of asphalt, concrete paved roads, and sidewalks not permeable to rainwater towards underground) [8–10] showed that, as in Figure 4, the ratio of the impervious area and water quality items were positively (+) correlated. Other alternatives may be the selection of priority management areas depending on the area where many complaints have occurred and the area preferred by local governments. However, in this case, it was difficult to quantitatively and equitably quantify problems which may have arisen.

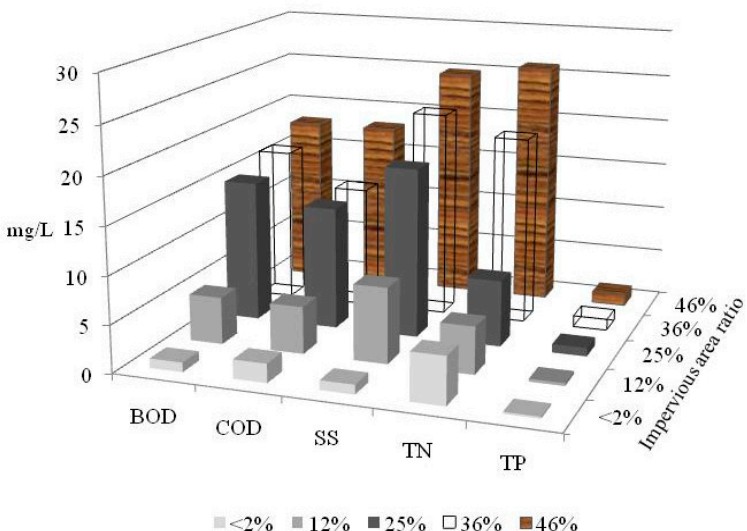

**Figure 4.** Water quality of the watershed by impervious area.

In Schueler et al. [8] and Choi et al. [11], as shown in Figure 5, the increase in impervious area ratio degraded the watershed condition. The increase in the impervious area ratio led to the increase in rainfall runoff and the amount of combined sewer overflows (CSOs), degrading water quality and watershed condition. Furthermore, if the impervious surface ratio is below 10%, the river bed becomes unstable, erosion is developed, and temperature increases, which together degrade habitable environments and lead to a worse hydro-ecology. As shown in Figure 5, if the impervious area ratio is over 25%, the water quality and watershed condition rapidly deteriorate, and NPS management is required. Therefore, the impervious area ratio can be used as an important index for NPS management region selection [12]. A study in South Korea [13] that focused on the total maximum daily load (TMDL) unit watershed in the Han River also showed that small watersheds with a 10–25% impervious area ratio failed to achieve good water quality.

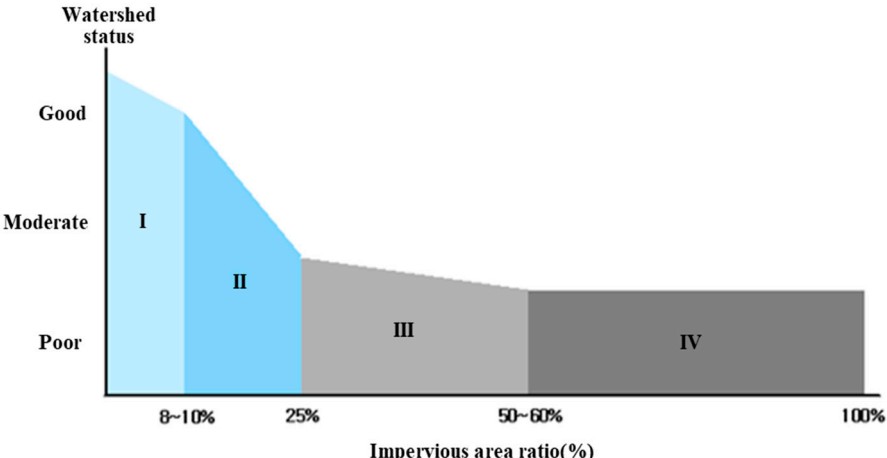

**Figure 5.** Watershed condition according to impervious area ratio.

Figure 6 shows the total phosphorus (TP) prioritized consolidation index estimation procedure for the determination of management region prioritization [14].

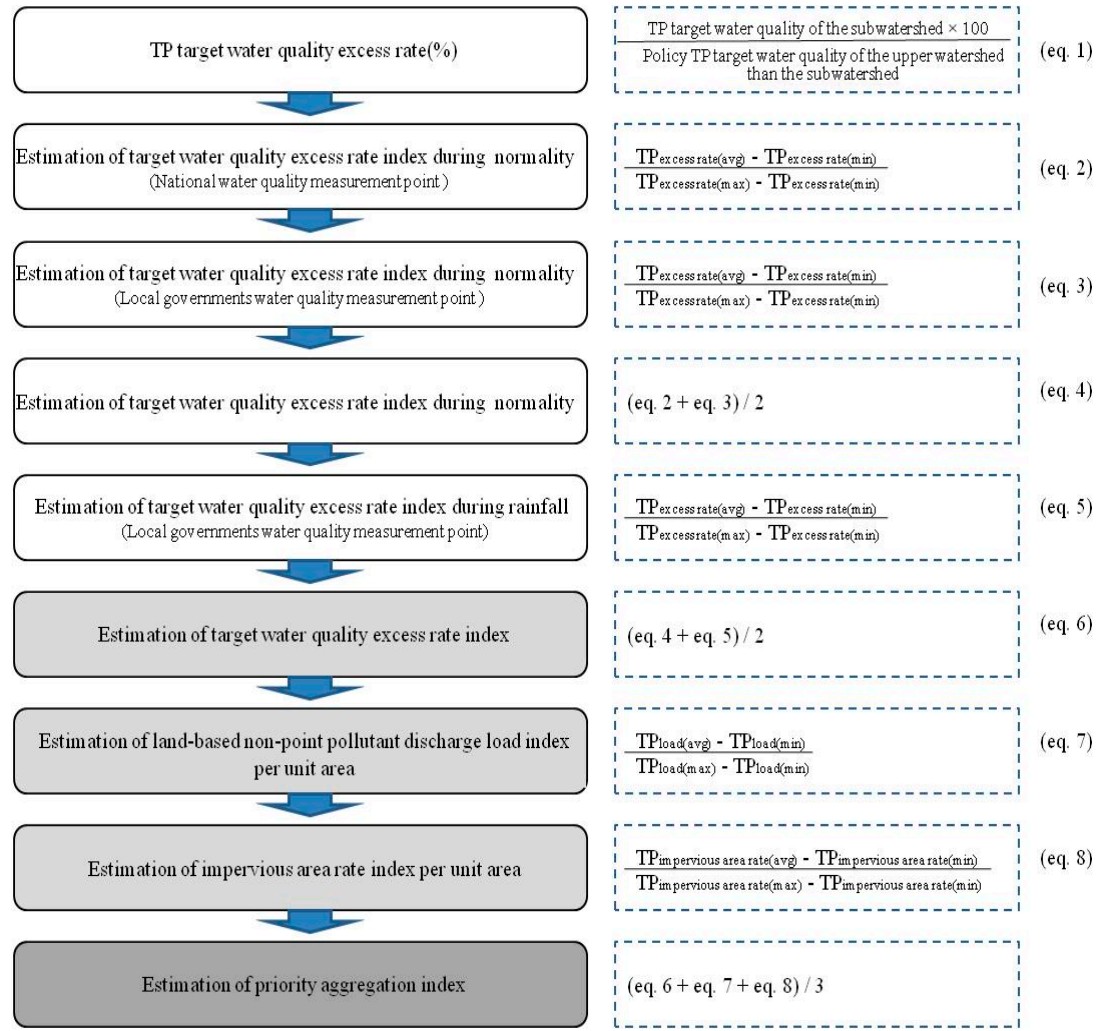

**Figure 6.** TP (total phosphorus) priority integration index calculation procedure.

The prioritized consolidation index is an average of the water quality target excess ratio index, the NPS load standardization index in unit area, and the impervious area ratio standardization index in unit area. The higher the prioritized consolidation index, the higher the priority order of the management region.

The water quality target excess ratio index is the average of the water quality target excess index in normal conditions (eq. 4) and during rainfall (eq. 6). The water quality target excess index in normal conditions was derived from the average (eq. 4) of the water quality target excess index at the water quality measurement site managed by the central government (eq. 2) and the water quality target excess index at the water quality measurement site managed by the local government (eq. 3) for fairness.

The NPS load standardization index and the impervious area ratio standardization index by unit area were determined by eq. 7 and eq. 8, respectively.

The prioritized cases in the watershed selected as NPS management regions (Figure 7) are shown in Table 2 and Figure 8 [15]. The data of the Figure 8 and Table 2 were used for the past three years (2016–2018) of Gimhae City TP water quality and flow rate data.

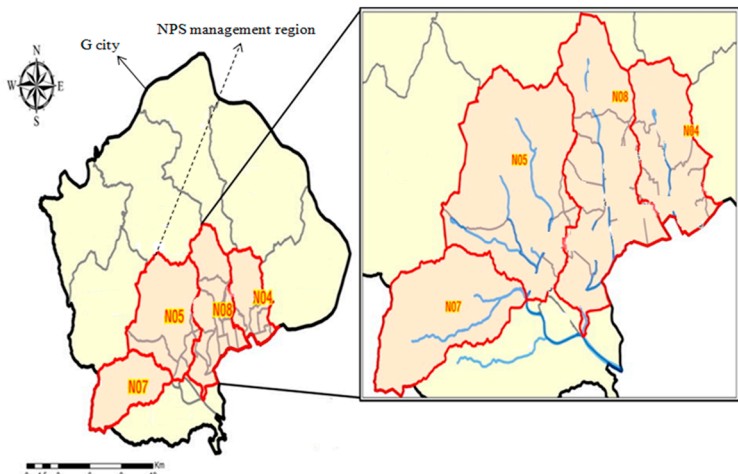

**Figure 7.** Prioritization methodology for TP management measures area.

**Table 2.** Priority index analysis results.

| Subwatershed | Water Quality Target Excess Rate Index | | | Non-point Source Pollutant Discharge Load Index of Land per Unit Area (4) | Impervious Area Rate Index per Unit Area (5) | Priority Consolidation Index ((3) + (4) + (5))/3 | Ranking |
| | Normal Condition (1) | Rainfall Condition (2) | Average (3) = ((1) + (2))/2 | | | | |
|---|---|---|---|---|---|---|---|
| N04 | 0.917 | 1.000 | 0.959 | 0.762 | 1.000 | 0.907 | 1 |
| N05 | 0.500 | 0.000 | 0.250 | 0.552 | 0.000 | 0.267 | 4 |
| N07 | 0.883 | 0.781 | 0.832 | 0.000 | 0.212 | 0.348 | 3 |
| N08 | 0.405 | 0.826 | 0.616 | 1.000 | 0.284 | 0.633 | 2 |

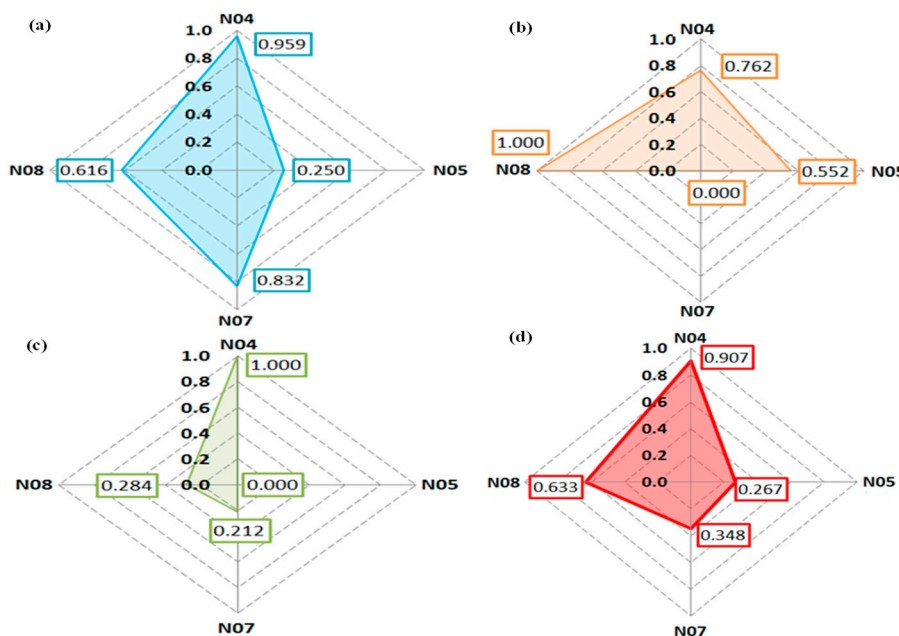

**Figure 8.** Priority index analysis diagram for each sub-watershed ((**a**): water quality target excess rate index, (**b**): non-point source pollutant discharge load index of land, (**c**): impervious area index, (**d**): priority consolidation index).

The concerned region consisted of four small watersheds: N04, N05, N07, and N08, and based on the procedure in Figure 5, the water quality target excess ratio index at the water quality measurement site, the water quality target excess ratio index at the water quality measurement site run by the local government, the water quality target excess ratio index in normal condition, the water quality target excess ratio index during raining, the NPS load standardization index, and impervious area ratio

standardization index by unit area were calculated. The prioritized consolidation index based on each factor index showed that N04 was 0.907, N05 was 0.267, N07 was 0.348, and N08 was 0.633. Accordingly, N04, with the highest consolidation index, would need NPS management most urgently, followed by N08, N07, and N05.

*3.2. Selection of Management Items*

Once the prioritization within the key management region was determined, the water quality pollutants were selected by the following procedure. First, the measured daily flow data (or data based on a model) and the cumulative flow data for the last five to ten years were used to create the FDC. Then, the management flow zone was determined. Since an NPS is closely connected to flow, the management flow zone could be determined by policy based on a high flow zone (excluding drastic flood periods, usually >5%), characteristics of watersheds, and the management objectives of the target region among others. The management flow rate section could be established from the moist condition between the low flows. However, in this case, in order to achieve the target standard, it was necessary to manage point source as well as non-point source. This will result in enormous costs and make it difficult to pursue policies focused on non-point pollution management.

Once the management flow zone was determined, the LDC was created based on the water quality standard of the target region (the water quality target managed by policy, TMDL water quality target) and the standard FDC. The measured load by pollutant was plotted on the standard LDC, and values that exceeded the standard LDC in the management flow zone became water quality pollutant items.

As shown in Figure 9, the measured load of BOD was below the standard LDC and did not require management, whereas that of TP exceeded the standard LDC and was set as a management item [16]. The data of the Figure 9 were used for the past 10 years (2008–2017) of Andong City flow rate, TP, and BOD.

*3.3. Selection of the Management Water Quality Target*

Once the management water quality target items were selected, the management water quality target value to be achieved during the NPS management plan needed to be determined. The management water quality target could be selected either by LDC or by concentration distribution.

3.3.1. LDC-Based Management Water Quality Target Selection

The LDC-based management water quality target selection was composed of the following five stages: creation of FDC for management flow zone selection, selection of a representative rainfall year for future simulation, establishment and estimation of the watershed model, selection of the water quality target, and creation of the water quality target LDC. Figure 10 shows the detailed explanation of the management water quality target [17].

In the FDC creation stage, the daily average flow data for over ten years in the management region's river were used to sort the daily average flow during the analysis period from the maximum to the minimum. Here, the X-axis was the percentage of the flow continuation zone, the percentage of the order/flow (high flow if the flow continuation zone is 0%; low flow if it is 100%), and the Y-axis was the daily average flow corresponding to the flow continuation zone. From the created FDC, the extreme flooding periods were excluded (5%), and within the general period (60%) zone (5–60%) that caused NPS pollution, the excess frequency characteristics of pollutants were considered to select the management flow zone.

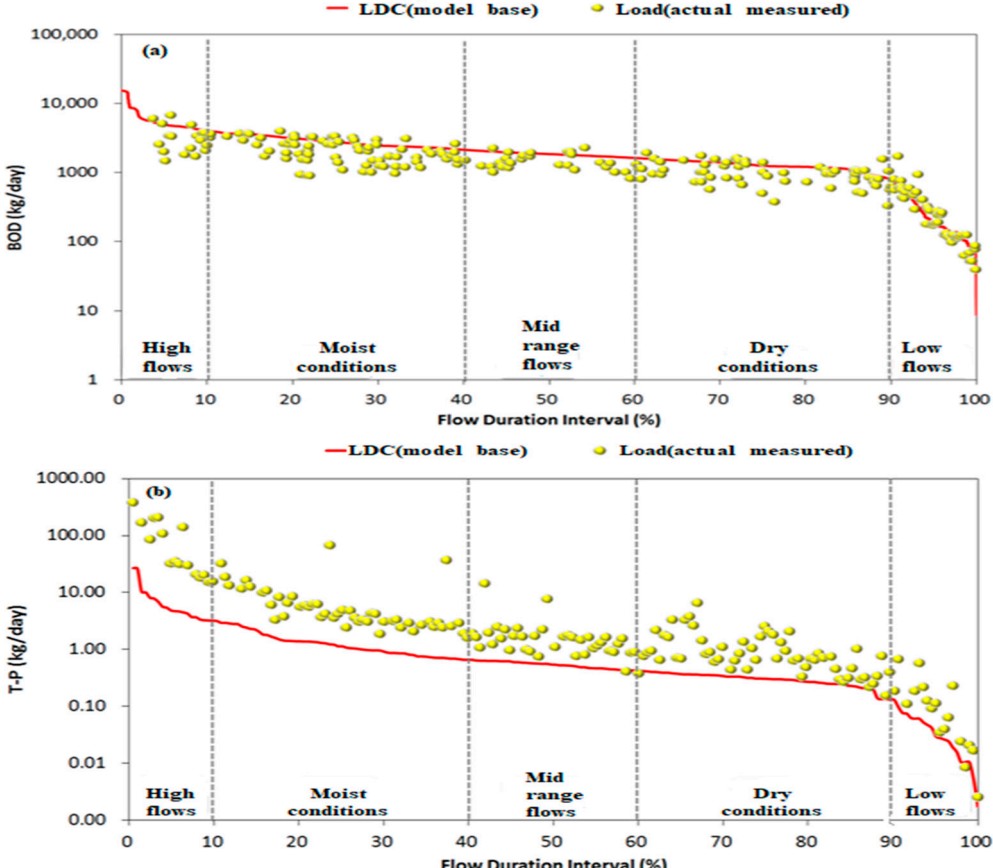

**Figure 9.** Example of management item selection ((**a**): BOD, (**b**): TP).

The selection of the representative rainfall year for future simulation was based on the acquired rainfall data for the last ten years to determine the total rainfall and seasonal average rainfall. The period in which the rainfall was most similar to the seasonal and the total average rainfall was set as the representative rainfall year.

In the watershed model establishment and estimation stage, geographic information system (GIS) data, the digital elevation model (DEM), land cover, soil map, and administrative map, etc., were used to establish a watershed model (Soil and Water Assessment Tool (SWAT), Hydrologic Simulation Program Fortran (HSPF)) suitable for the characteristics of the management region.

Using these factors, land slope and land use type could be analyzed to easily identify the distribution of non-point pollutants and the discharge characteristics of non-point pollutants. Another alternative to accurately diagnose the water quality effect by NPS is to measure the actual water quality at a point that can represent the water quality characteristics of the area where non-point pollution management is needed. However, it is difficult to predict future water quality when measuring the water quality directly. Therefore, it is essential to use the watershed model to set long-term goals for the management of non-point pollution sources and to carry out policies.

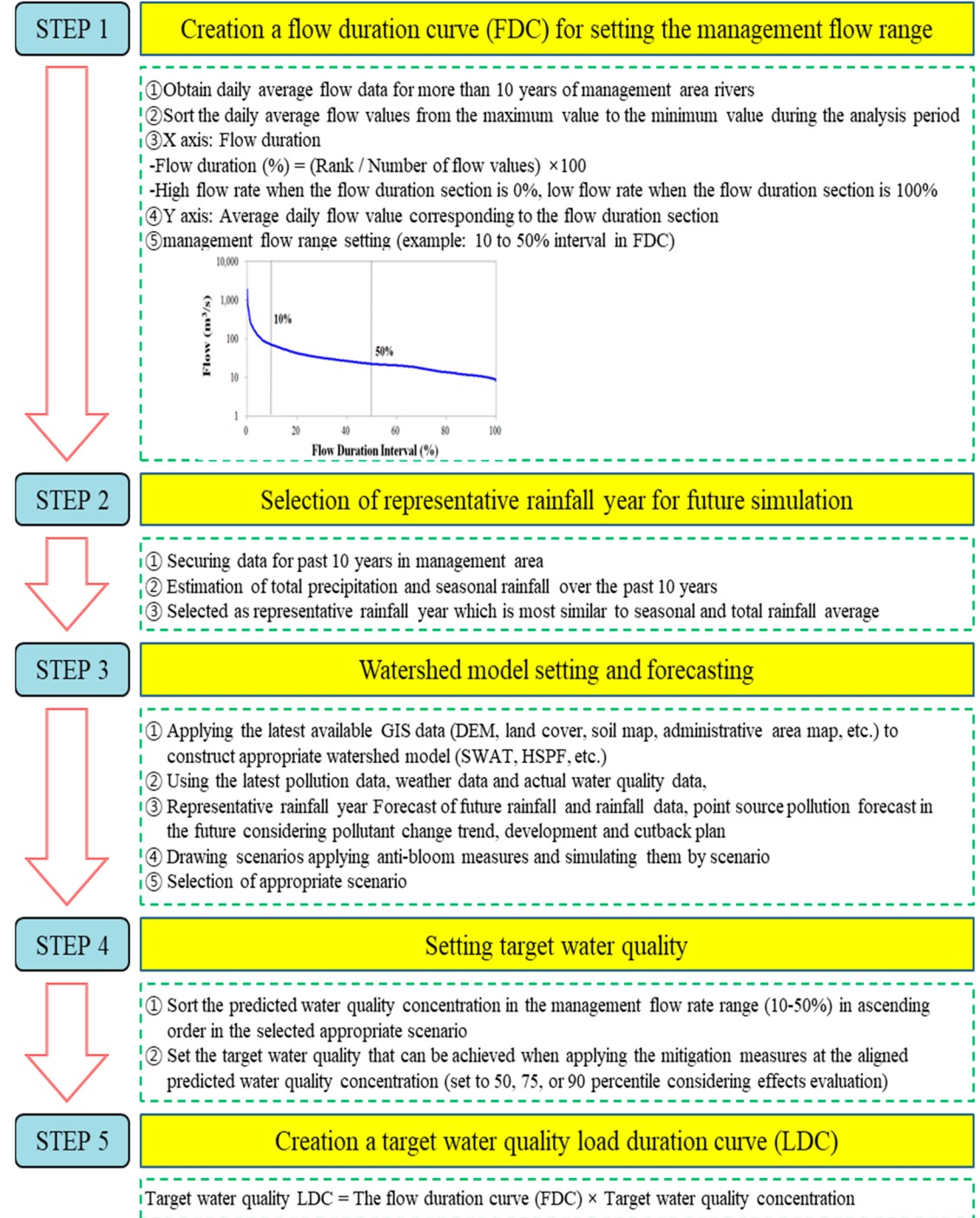

**Figure 10.** Setting the management water quality target and creating the load duration curve (LDC).

The most up-to-date pollutant source data, meteorological data, and water quality measurement data were used to revise and verify the watershed model and improve the completion level of the established model. The climate and rainfall data in the representative rainfall year and the future prospect of point sources (pollution source trends, development and reduction plans, etc.) were considered to estimate the water quality and load when an NPS management plan is not established. Furthermore, scenarios with an NPS management plan were developed and the future water quality and load by each scenario were simulated to determine the optimal scenario. If the appropriate scenario is not determined, it will be difficult to predict which non-point pollution reduction facilities

are effective in the NPS management area, how much it costs, and whether the target water quality can be achieved within the management period.

Once the optimal scenario was determined, the specific level of the estimated load (50th, 75th, 90th percentile) was set as the water quality target, while the water quality target LDC was not exceeded after the NPS reduction plan in the management flow zone.

Figure 11 shows a case of a water quality target based on LDC when TP was the target item within the management flow zone between 5% and 25% [18]. Within 5% and 25% of the management flow zone, 0.34 mg/L, the level at which the 95th percentile of the TP load estimated based on the optimal scenario did not exceed the water quality target LDC, was set as the water quality target. The data of Figure 11 were used for the flow rate and TP data of seven cities located in Jeollabuk-do for the past 10 years (2008–2017).

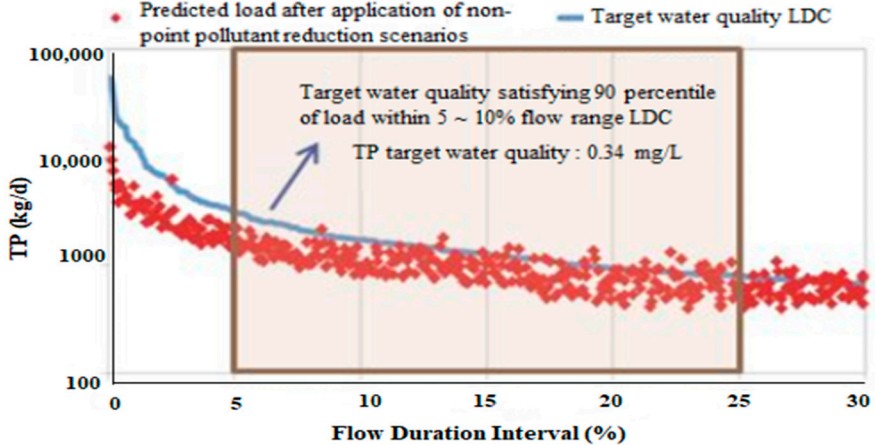

**Figure 11.** Example of setting water quality target using LDC.

3.3.2. Selection of the Management Water Quality Target Based on Concentration Distribution

The selection of the management water quality target based on concentration distribution used a method almost identical to the one that used LDC in Section 3.3.1. However, the method based on concentration distribution determined the optimal reduction scenario based on the watershed model, which was then used to reduce NPS pollutants, and with the estimated water quality at the target site, the management water quality target was determined. In other words, the water quality target was determined in the level that satisfied the suitable level of the estimated concentration (for example, 95th percentile) when the NPS pollutant in the management flow zone was reduced based on the optimal scenario.

Shown in Figure 12 is an example of the selection of the management water quality target that used concentration distribution [18]. Within 5–10% of the management flow zone, 108.4 mg/L, which satisfies the 95th percentile in the SS water quality estimated by the optimal scenario, was determined to be the water quality target.

The data of Figure 12 were used for the flow rate and SS data of seven cities located in Jeollabuk-do for the past 10 years (2008–2017).

*3.4. Evaluation of the Management Water Quality Target*

After completion of the NPS management plan, the feasibility and performance of the plan need to be reviewed by evaluating whether each item of the water quality target has been achieved. The evaluation method was identical to the method used in setting the water quality target: the water quality target achievement level at the measurement site (for the last three years' water quality measurement data in the management flow zone) and the LDC achievement ratio (%) (the actual measured load for the last three years in the management flow zone (the measured water quality X flow)). The LDC

achievement ratio could be determined based on policy by considering the water quality and pollutant load in the concerned region, and it generally used 75%.

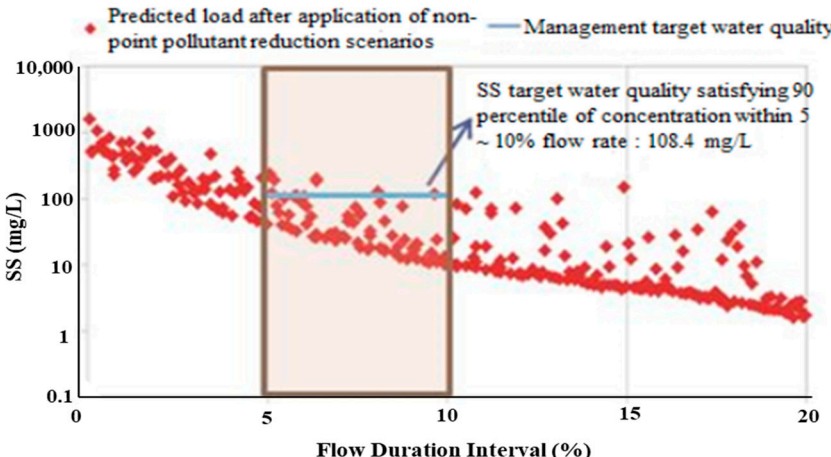

**Figure 12.** Example of settingwater quality target using concentration distribution.

Figure 13 shows an example of the management target evaluation for the total phosphorus where the target achievement ratio was 75%, the management flow zone was 10 to 50%, and the measured load was 100 [17]. First, the measured load was plotted on the LDC with the 75th percentile optimal value as the water quality target to determine what percentages of the load were distributed in zone A (below 75th percentile LDC) and in zone B (over 75th percentile LDC). If zone A had a cover of 75%, the target had been achieved.

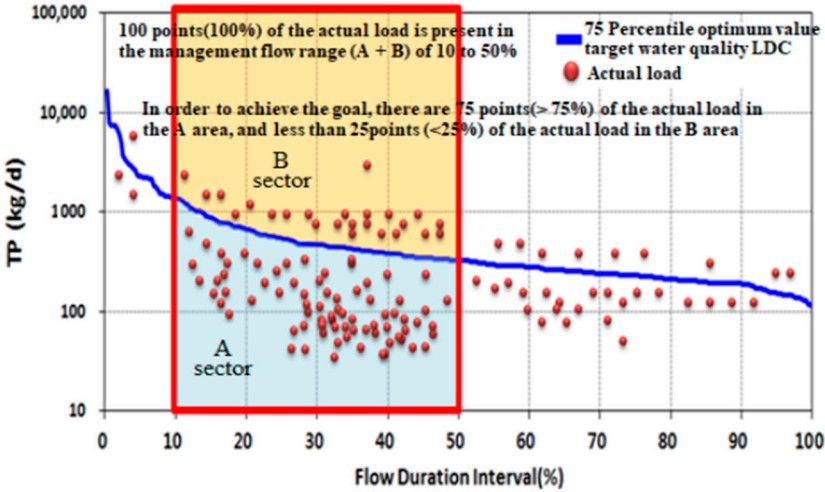

**Figure 13.** Example of management water quality target evaluation.

Figure 13 used flow rate and TP data of the upper stream area of Soyang Lake (Mandae, Gaa, Jaun district, Switzerland).

## 4. Conclusions

To manage NPSs, it is necessary to implement processes and a selection method for key components in the NPS management plan that can be applied in any region for the management of NPSs.

For the selection of NPS management items and regions, water quality items affected by NPSs should be determined. Water quality simulation for the water quality items in the target region after

the reduction of NPS load and the water quality target in the region as a policy were compared to analyze the frequency of excess, and in the case of a high excess ratio during rain when NPS-based water pollution occurred, the concerned water quality items should be designated as management items, and said region should be designated as an NPS management region.

Currently, there are 14 NPS management regions, 11 of which have an NPS management plan.

Regarding water quality target items, SS, which causes turbidity by muddy water, was the largest in five regions; TP, which causes algae creation, was present in four regions; and BOD was high in two regions.

The management flow zone was determined within the scope of a high flow zone (5–60%) by considering the frequency of the excess of the management water quality target items in the mid-level region's water quality target, among others.

When determining the priority locations within the management plan region, key factors of NPSs, such as the water quality target excess ratio, the point discharge load of land, and impervious area ratio, were indexed quantitatively and evaluated to allow for an efficient NPS management.

The management water quality target can be determined by LDC or concentration distribution. The LDC-based management water quality target was determined by the following procedure: the creation of FDC for setting the management flow zone; the selection of the representative rainfall year for future simulation; establishment and estimation of the watershed model; the selection of the water quality target; and the creation of the water quality target LDC, and the water quality target is set within the level that does not exceed the water quality target LDC after the NPS load reduction plan in the management flow zone.

The concentration distribution-based method was almost identical to the LDC method, but it selected the optimal scenario using the watershed model, and the management water quality target was determined based on the water quality estimated at the target site after the reduction of NPS pollutants based on the identified scenario.

In order to manage non-point pollutants, reflecting more effective and regional characteristics, various studies on factors that should be considered besides the core elements presented in this manuscript are required. In addition, it is necessary to study and evaluate the effects of these factors on the performance evaluation of non-point pollution management measures.

**Author Contributions:** Conceptualization, D.S.S., J.K.L., J.R.; methodology, J.R.; data curation, J.R., J.H.P.; writing—original draft preparation, J.H.P.; writing—review and editing, J.H.P.; supervision, D.S.S., J.K.L.

**Funding:** This research was supported by the National Institute of Environmental Research under Grant Nos. 1900-1946-303.

**Acknowledgments:** This research was conducted as a regular research activity in National Institute of Environmental Research (NIER-RP2018-248, etc.).

**Conflicts of Interest:** The authors declare no conflicts of interest.

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
