# Peer review of "Methodology for Determining the Key Factors for Non-Point Source Management"

_water, doi:10.3390/w11071381_

Round 1

Reviewer 1 Report

The study introduces steps and factors in establishing a NPS management plan, including the designation of NPS management regions, the selection method for key management areas, the selection method of management items and the water quality target, and an evaluation method of the water quality target. This is meaningful for NPS management for a country. However, the structure and layout of the article are a bit confusing and some parts are unclear. In addition, the conclusion is not concentrated and does not show the significance of this study. There are some questions below:  

1.          In the abstract, a lot of process in establishing a NPS management plan were described, but which of these were carried out by this study was not clear.

2.          I suggest that some materials be put in the second part, such as table 1 and figure 2. And some descriptions of the current situation in the third part might be better to put in the first or second part.

3.          The method mentioned in this article has many steps, so it will be helpful to add a flowchart of the whole method.

4.          Please describe the data source.

5.          Line 106. The suitable watershed model here refers to? What kind of model was used in the example?

6.          Line 129. There are many factors influencing NPSs, why do you choose these three factors? Are these indicators representative?

7.          The resolution of pictures is low, can you replace them by clearer pictures?

8.          Line 189-192 “the management flow zone can be determined by policy based on a high flow zone, characteristics of watersheds, and the management objectives ” However, according to your description, the management flow zone seems to be determined only by high flow zone.

9.          The selection of management items was based on LDC and the selection of Management Water Quality Target was based on LDC or Concentration Distribution. Is there any difference LDC in 3.2 and 3.3? Why do you repeat this method?

10.      What do you mean by the abbreviation of “LCD”? whether it should be LDC?

11.      Line 278 “The ratio of NPSs in South Korea based on the BOD load is about 70% ” this did not conclude by this study, so it might be better to not mention it in the conclusion. 

Author Response

Response to Reviewer 1 Comments

Point 1: In the abstract, a lot of process in establishing a NPS management plan were described, but which of these were carried out by this study was not clear.

Response 1: Considering your comments, I have specifically described what we have done in this study And I put it at the end of the abstract.

Point 2: I suggest that some materials be put in the second part, such as table 1 and figure 2. And some descriptions of the current situation in the third part might be better to put in the first or second part.

Response 2: Reflecting your opinion, I reassigned the contents of Table 1 and Figure 2 to the second part. Thank you for your suggestion.

Point 3: The method mentioned in this article has many steps, so it will be helpful to add a flowchart of the whole method.

Response 3: We added the flowchart of the whole method described in this paper to make it easier for readers to understand.

Point 4: Please describe the data source.

Response 4: The data source of the data used in this paper is described.

Point 5: Line 106. The suitable watershed model here refers to? What kind of model was used in the example?

Response 5: We used various models (e.g. SWAT, HSPF, etc.) according to the NPS management regions. It was explained in the fourth paragraph of the section 3.3.1 and the step 3 of Figure 10.

Point 6: Line 129. There are many factors influencing NPSs, why do you choose these three factors? Are these indicators representative?

Response 6: In general, the factors that indicate the impact of nonpoint sources can vary widely. However, we have presented three major factors in this study. The reason is that the Ministry of Environment supports 70% of the total budget when it is designated as a nonpoint pollution management area in Korea. Therefore, many municipalities are applying for non-point pollution control areas, but due to budget limitations, Ministry of Environment can not support all municipalities. Therefore, the three most important factors in selecting non-point source pollution management areas in the Ministry of Environment are the factors presented in this study.

Point 7: The resolution of pictures is low, can you replace them by clearer pictures?

Response 7: Reflecting your comments, we revised the picture with low resolution to a picture with high resolution.

Point 8: Line 189-192 “the management flow zone can be determined by policy based on a high flow zone, characteristics of watersheds, and the management objectives …” However, according to your description, the management flow zone seems to be determined only by high flow zone.

Response 8: One of the most important factors in setting the management flow zone should be considered in the rainfall flow part that can cause nonpoint pollution.

However, when setting the management flow zone, various factors such as characteristics of watersheds, and the management objectives are considered. Therefore, the management flow zone is not set only in the High flow zone. The management flow zone is set to a case by case depending on the region. This is also shown in the table 1. Depending on the region, the management flow zone may be set in the "high flow zone", or it may be set in the "high flow zone ~ mid range flow" or in the "moist condition ~ mid range flow". However, this paper does not describe all the cases of considerations because the cases vary depending on the region.

Point 9: The selection of management items was based on LDC and the selection of Management Water Quality Target was based on LDC or Concentration Distribution. Is there any difference LDC in 3.2 and 3.3? Why do you repeat this method?

Response 9: The LDCs in section 3.2 and 3.3 have somewhat different meanings.And it's not used repeatedly.
The LDC for the establishment of the management items in section 3.2 means the load amount derived by multiplying the water quality target managed by policy (e.g., TMDL, etc.) by the flow rate.
The LDC for the establishment of the management water quality target in section 3.3 means the load amount derived by multiplying the flow rate to the water quality which is reduced by scenario by the planned nonpoint pollution reduction facility in the future.

Point 10: What do you mean by the abbreviation of “LCD”? whether it should be LDC?

Response 10: I'm deeply sorry. It's our errata. It means "LDC".

Point 11: Line 278 “The ratio of NPSs in South Korea based on the BOD load is about 70% …” this did not conclude by this study, so it might be better to not mention it in the conclusion.

Response 11: I have excluded the content from the conclusion, reflecting your opinion. Thank you so much.

Reviewer 2 Report

Manuscript no. water-530107

Title: "Methodology for determining the key factors for non-point source management"

The manuscript describes the current situation of the designation of NPS regions in South Korea.

The Introduction section offers a sufficient background, and the Methods are clearly described, although my research profile (more relevant to research topics related to water quality) does not allow me to analyze the specific aspects of the adopted methodology.

On the contrary, I believe that the Results section and the Conclusions section could be implemented.

However, I believe that the work could be of interest to readers, even though it is mainly related to the reality of North Korea. For these reasons, I have minor suggestions for the authors.

In particular:

1) In the Abstract, the Authors use abbreviations (see line 18 - TMDL, and line 21 - SS) that are not previously declared in the text. All abbreviations must be declared in extenso, and this problem is present throughout the whole text, not only in the abstract. For example, see the lines: 105 (SS?); 138 (CSO?); 150 (TP?). This problem makes the manuscript really difficult to read. The Authors are strongly urged to use the abbreviations sparingly and to make them understandable before their use in the text.

2) Lines 39-41: to which research do the Authors refer? A personal search? This part must be implemented to provide clarity to the readers.

3) Line 54: bring the percentage back to one decimal place.

4) Figure no. 1 and figure no. 10 have low quality and definition. I recommend improving the quality of these images.

5) The Conclusions section is not exhaustive, as it contains simple discussions of the results. The Authors must review this section, implementing the possible applications of their findings and future research developments. It would also be advisable to make comparisons and references to the international panorama.

6) Finally, I suggest specifying much more clearly the purpose of the research at the end of the Introduction section.

Author Response

Response to Reviewer 2 Comments

Point 1 : In the Abstract, the Authors use abbreviations (see line 18 - TMDL, and line 21 - SS) that are not previously declared in the text. All abbreviations must be declared in extenso, and this problem is present throughout the whole text, not only in the abstract. For example, see the lines: 105 (SS?); 138 (CSO?); 150 (TP?). This problem makes the manuscript really difficult to read. The Authors are strongly urged to use the abbreviations sparingly and to make them understandable before their use in the text.

Response 1: If we used abbreviations on paper, we added a description of abbreviations in the text.

Point 2 : Lines 39-41: to which research do the Authors refer? A personal search? This part must be implemented to provide clarity to the readers.

Response 2: The frequency of use of non-point pollution terms was personally surveyed by the authors and explained that on the paper.

Point 3 : Line 54: bring the percentage back to one decimal place.

Response 3: Reflecting your comments, we matched the percentage to one decimal point.

Point 4 : Figure no. 1 and figure no. 10 have low quality and definition. I recommend improving the quality of these images.

Response 4: Reflecting your comments, we revised the picture with low resolution to a picture with high resolution.

Point 5 : The Conclusions section is not exhaustive, as it contains simple discussions of the results. The Authors must review this section, implementing the possible applications of their findings and future research developments. It would also be advisable to make comparisons and references to the international panorama.

Response 5: As you commented, I reconstructed the conclusion of the manuscript and modified the paper.

Point 6 : Finally, I suggest specifying much more clearly the purpose of the research at the end of the Introduction section.

Response 6: In the introduction, we explained the purpose of the study more specifically.

Round 2

Reviewer 1 Report

All my comments have been raised. I suggest the acception of this paper.

Author Response

There was no additional comment, so we did not create an answer.
Thank you for reviewing our paper.